# Diurnal Monitoring of Moisture Content of Scots Pine and Small-Leaved Lime Trunks Using Ground Penetrating Radar (GPR) and Increment Cores

**Maria Sudakova** [1,2,*], **Eugenia Terentieva** [1], **Alexey Kalashnikov** [3], **Ivan Seregin** [4,5] **and Alexey Yaroslavtsev** [4,5]

1. Geology Faculty, Seismic Department, MSU Lomonosov, GSP-1, 1 Leninskiye Gory, 119991 Moscow, Russia
2. Earth Cryosphere Institute, Tyumen Scientific Centre SB RAS, Maligyna 86, 625000 Tyumen, Russia
3. Department of Landscape Design and Sustainable Ecosystems, Agrarian-Technological Institute, Moscow State University of Civil Engineering, 26, Yaroslavskoye Shosse, 129337 Moscow, Russia
4. Department of Ecology, Russian State Agrarian University-Moscow Timiryazev Agricultural Academy, Timiryazevskaya St., 49, 127550 Moscow, Russia
5. Agrarian and Technological Institute, Peoples' Friendship University of Russia, Miklukho-Maklaya Str., 6, 117198 Moscow, Russia
* Correspondence: m.s.sudakova@yandex.ru

**Abstract:** Ground penetrating radar is non-invasive technology suitable for mapping moisture content variations since it shows high sensitivity to changes in water saturation. In this work we used a GPR tomography approach to estimate moisture content within two small-leaved lime (*Tilia cordata*) and two Scots pine (*Pinus sylvestris*) trunks. Additional information was derived using the method of GPR zero-offset. GPR data was collected in Moscow (diurnal monitoring in September 2021) using a shielded GPR antenna working at 1500 MHz. Moisture values derived from GPR data were compared with the values obtained directly by measuring sampled wood cores gravimetrically. A good agreement was observed between GPR-derived moisture content and core sample-derived values. Notwithstanding GPR-derived moisture content is about two times higher than core sample-derived values, a strong linear relation with a determination coefficient more than 0.8 is observed. Diurnal monitoring did not reveal any significant changes in moisture content inside the trunks. It can be concluded that the period of early autumn in the Moscow region is characterized by a constant moisture content of the small-leaved lime trunk during the day.

**Keywords:** non-destructive testing; tree stability; wood decay; wood moisture content; trees monitoring





## 1. Introduction

Stem water content is an important factor controlling a living tree's health and is variable in the interaction processes between soil, stem and atmosphere. The detection of the early symptoms of tree decay represents a major challenge for the identification of tree diseases or tree hollows. High moisture content can initiate wood fungal decay [1]. Temporal and spatial moisture content variation in response to environmental, seasonal factors and stages of tree growth are difficult to measure [2–4].

In most tree species, xylem consists of two functional parts: sapwood and heartwood. The sapwood acts as the transportation agent for nutrients and moisture from the tree roots to the leaves. As long as a newer layer of sapwood is formed, the inner layer becomes inactive and transforms into heartwood. Sapwood has been extensively studied; at the same time, the details of flow exchange between sapwood and heartwood are less elaborated [5]. Moisture content (MC) variation may be indicative of sapwood to heartwood area ratios and allows the production of tree transpiration estimates.

Stem water content can be estimated using non-invasive and invasive methods. Invasive techniques, some of which require drilling, result in wood tissue exposure to fungi

invasion and spread of infection to healthy wood [4,6,7]. From a broader perspective, trees live for decades or longer, so destructive sampling cannot be considered a sustainable method for repeatable (time-lapse) observations if the objective is to preserve the tree health.

Among non-invasive methods of sap flow measurement, the most widely used is the electrical resistivity method [4,8,9]. A steep increase in electrical resistivity at the sapwood–heartwood boundary to the pith is observed which is accounted by the properties contrast in sapwood (wet) and heartwood (dry) [9]. During heartwood formation, moisture content decreases before extractive contents reach levels visible by staining [10]. However, this observation may be due to other factors such as temperature, moisture content, wood structure and chemistry concentration [11–13].

Ground penetrating radar (GPR) has been increasingly used in forestry given its flexibility, rapidity of data collection and cost-efficiency. GPR techniques (among which tomographic inversion) can be used to estimate water content due to the sensitivity of electromagnetic velocity to water content and has a potential to accurately detect decay in living trees [14,15]. GPR tomographic inversion is based on electromagnetic (EM) waves transmitted through the medium. The travel time of EM waves is controlled mainly by the dielectric properties of the medium which determine the corresponding EM wave velocity. Travel times are used as input data, and the output tomogram characterizes the velocity field [16]. Processing and tomographic techniques, elaborated for the seismic industry, are nowadays being applied to GPR data and to solve multiple tasks including characterization of shallow subsurface features (wet zones, cavities, hydrologic infiltration and much more).

From a perspective of live tree trunk observations, GPR tomography was successfully applied to the search and estimation method for internal defects in live tree trunks [15] and quantitative estimation of moisture content [17].

In this study, we combine GPR in tomographic and zero-offset (ZO) modification with direct measurements (coring) for estimation of sapwood area and diurnal moisture content variation in various live tree species: coniferous Scots pines and deciduous small-leaved limes. Differentiation between sapwood and heartwood is based on a steep decline in wood moisture [18,19].

## 2. Materials and Methods

Investigation was performed at Timiryazev urban forest (55.8178 N, 37.5565 E), which belongs to the few conservation infrastructure sites in Moscow city that have preserved the natural soil cover, dominated by sod-podzolic soils with varying levels of organogenic, humus-accumulative, podzolic and transitional horizons with varying degrees of hydromorphism and gleying, which are typical of the background southern taiga ecosystems of central European Russia [20]. The mean annual temperature was 5.8 °C and the mean annual precipitation was 708 mm. A position of the investigation site in Timiryazev urban forest in Moscow city, Russia is shown on Figure 1.

The Forest Experimental Dacha (FED) of the Timiryazev Academy is one of the oldest research and educational institutions in forestry in Russia based on Timiryazev urban forest. In 1862, A.R. Vargas de Bedemar was invited to set up the FED and compiled the first growth tables for oak, lime and birch stands [21].

The forest stands of FED consist of stands of natural and artificial origin, presented in almost equal proportions. The average stand age is about 100 years [20].

The selection of the key area and representative species was determined from historical data and forest characteristics. A reforestation experiment led by A.R. Vargas de Bedemar was carried out at the research site area. Conifers were planted and sown, but from 1938–1939 drought resulted in the death of the experiment. Subsequently, a small-leaved lime undergrowth was recorded at this site. According to the FED forest inventory completed in 2009, the dominant species was Scots pine (Pinus sylvestris), and small-leaved lime stands grow on an area of 5.03 ha with a predominance of mature trees (4.55 ha) [21].

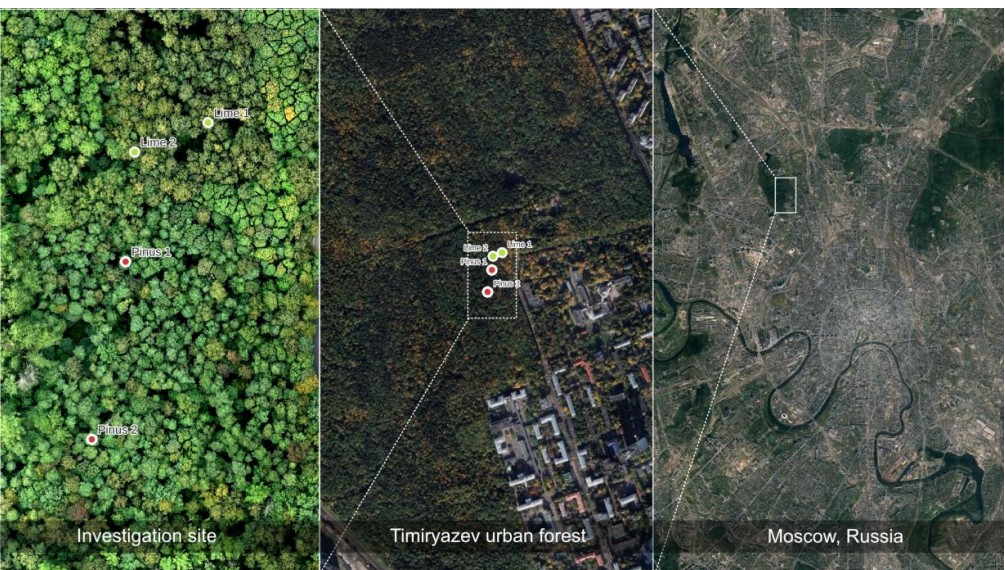

**Figure 1.** Position of the investigation site in Timiryazev urban forest in Moscow city, Russia.

One of investigated species was Scots pine (Pinus sylvestris) which has the widest geographic range of all tree species in Eurasia. Most conifers are known for the higher moisture content of sapwood compared to heartwood [9]. The other examined species was Tilia cordata (small-leaved lime tree) which is native throughout most of the Northern Hemisphere [22]. Measurements were carried out on a pair of trees of each species for more robust results.

Trees were chosen according to visual tree assessment. There were no visible trunk defects, dry branches or dry or discolored leaves; all trees' bark was uniform. The trees on the examined plot were about 160 years old according to historical data [21] and had a circumference range of 128–134 cm. The circumferences of trees were small-leaved lime 1—127 cm, small-leaved lime 2—150 cm, Scots pine 1—136 cm, Scots pine 2—157 cm. All measurements were taken at breast height (130 cm).

The monitoring of the moisture content of tree trunks was conducted using ground penetrating radar (GPR) and gravimetrical analysis of wood core samples. The samples were taken from the same trees were the GPR measurements were conducted. The GPR tomography process is shown in Figure 2.

### 2.1. GPR Method

GPR measurements were made around the trunk to obtain one slice at a height of about 130 cm from the ground level. GPR was carried out using the zero-offset method (ZO) and tomography technique. The typical mode of GPR operation is the common-offset mode where the receiver and transmitter are maintained at a fixed distance and moved along a line. If the distance between the two antennas is negligible, this is called zero-offset data (ZO) [23].

The measurements were carried out 3 times during daylight hours on 12 September 2021 in the morning (9 a.m.), afternoon (at noon) and evening (5 p.m.). In both cases a GPR Zond-12e with shielded bowtie antennas with a central frequency in the air of 1500 MHz was used in data acquisition.

ZO data processing included interpolation between marks, high-frequency filtering, amplitude correction for spherical divergence and conversion to polar coordinates. The distance between the marks was 10 cm.

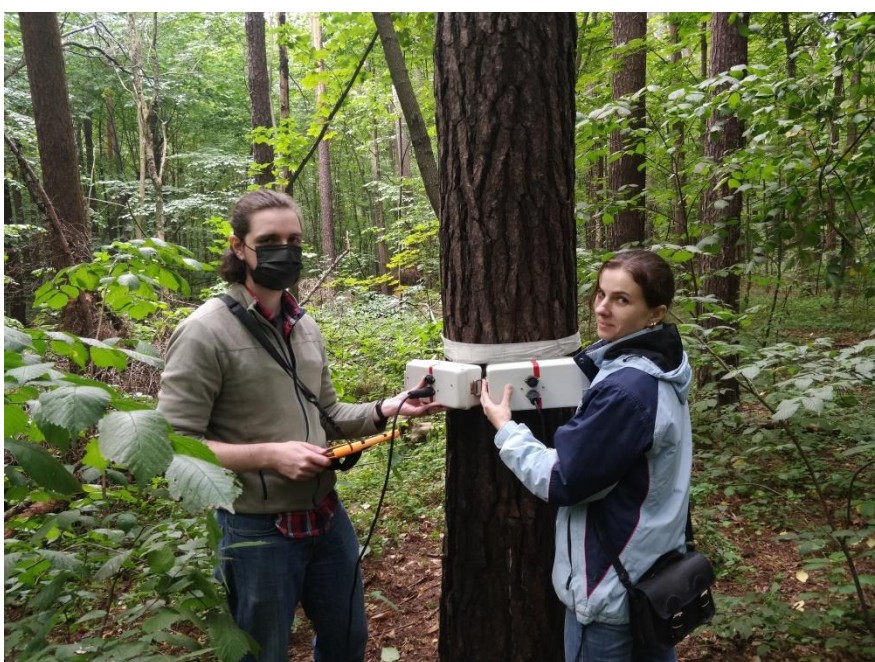

**Figure 2.** GPR tomography data acquisition.

　　To collect tomography data, the source and receiver points were located in a horizontal plane along the perimeter of the trunk. The transmitting antenna was placed against the trunk of a tree and the receiver moved circumferentially to acquire a complete B-scan of the selected trunk slice. The influence of induction and conduction currents was eliminated when minimal distance between transmitting and the receiving antennas achieved 20 cm. Co-polarized broadside antenna orientation was applied [23]. The step between the radiation points was 10 cm. The work was carried out in continuous mode and field trace spacing was about 1 mm. During processing, the trace spacing was increased to 1 cm. The traces calibration was performed using a tape measure and markers placed in 10 cm increments. As a result, for a slice of the trunk, several thousands of travel time samples were acquired. Data processing involved traces interpolation and delay removal based on direct arrival. After processing, the direct arrival was picked. The picking was performed on a signal maximum, and then shifted to the first break time. The travel times were used to solve the inverse problem of ray tomography. The tree trunk was approximated by cylinder shape. Velocity sections were produced in GeoTomCG software; the grid cell size was 2 cm$^2$.

Moisture Content Calculation

　　Moisture content of wood (MC) is defined as the weight of water in wood expressed as a fraction, usually a percentage, of the weight of oven-dry wood [24].

$$MC = (Wtest - Wovendry)/(Wovendry) * 100\% \tag{1}$$

where *Wtest*: weight of test wood; *Wovendry*: weight of oven dry wood.

　　According to Equation (1), MC ranges from 0% for oven-dry wood and may exceed 200% of the weight of wood substance. In softwoods, the moisture content of sapwood is usually greater than that of heartwood [25]. The only direct method to determine MC in wood material is to measure the water content in a wood sample and the weight of an oven-dry sample.

　　Laboratory measurements of the electrophysical properties of wood at different frequencies [26,27] showed that water is the main component that determines the electrophysical properties of wood, including the dielectric permittivity.

The results of travel time tomography—the velocities of electromagnetic waves—were recalculated into moisture values according to an empirical formula, selected for wood with a density of 500 g/cm$^3$. In our work, to estimate the moisture content in trees, we used the correlation relationship between the real part of the dielectric permittivity versus moisture content.

The dielectric permittivity versus MC for various substances and media, including wood [27–29] at GPR frequencies can be expressed as follows (Equation (2)):

$$\varepsilon = aW^3 + bW^2 + cW + d \tag{2}$$

where *a*, *b*, *c* and *d* are coefficients selected individually in each case. The relationship is a third-order polynomial regression similar to the Topp approach [30]. The formula derivation is given in detail in the article [17]. Similar third-order polynomial calibration curves have been reported for water volume in other species [31–34]. Three correlation relationships (2) were constructed for 20 °C, the frequency of 500 MHz and densities 0.5 using the least squares method. The density value was taken from the paper [26].

### 2.2. Core Sampling

Wood cores were sampled in the morning (about 9:00 a.m.) and in the afternoon (about 3:00 p.m.) at breast height (130 cm from ground level) with a Haglöf increment borer from the southern and northern side of the tree and were processed.

All cores were released from the borer and measured in field, wrapped in stretch film, placed in sliding channel storage bags (Ziploc) and kept in a refrigerator at −20 °C. The obtained samples were measured in diameter and length after deep freezing; the diameter and length of the cores were measured again to detect changes in size. The length and diameter of the sections were measured using a Matrix electronic caliper with an error of 0.02 mm. Then, the first 2 cm were divided into 4 segments (5 mm) and the rest of each core was cut in parts 1 cm long. Subsamples (cuts) obtained were given unique IDs and weighed with a CAS CUX 620H class II laboratory scale with an accuracy of 0.001 g. Then, subsamples were placed in a drying chamber for 2 h at 80 °C and weighed afterwards. Drying and weighing cycles were repeated with subsamples until stabilization of their weight. Assuming that weight loss could happen only due to loss of the water, gravimetric water content was calculated.

## 3. Results

Figures 3–6 show examples of GPR data. Figures 3a, 4a, 5a and 6a show examples of GPR tomography data. A blue line on the figure demonstrates the first arrivals—the wave which propagates between the source and the receiver through the air. Its hodograph is a straight line. The maximum arrival time corresponds to the maximum distance between the source and the receiver, equal to the diameter of the trunk. The direct wave is distinguished in later arrivals: it travels between the source and receiver through the trunk of a small-leaved lime tree (green line on figure). The GPR tomography data are given before the delay removal procedure.

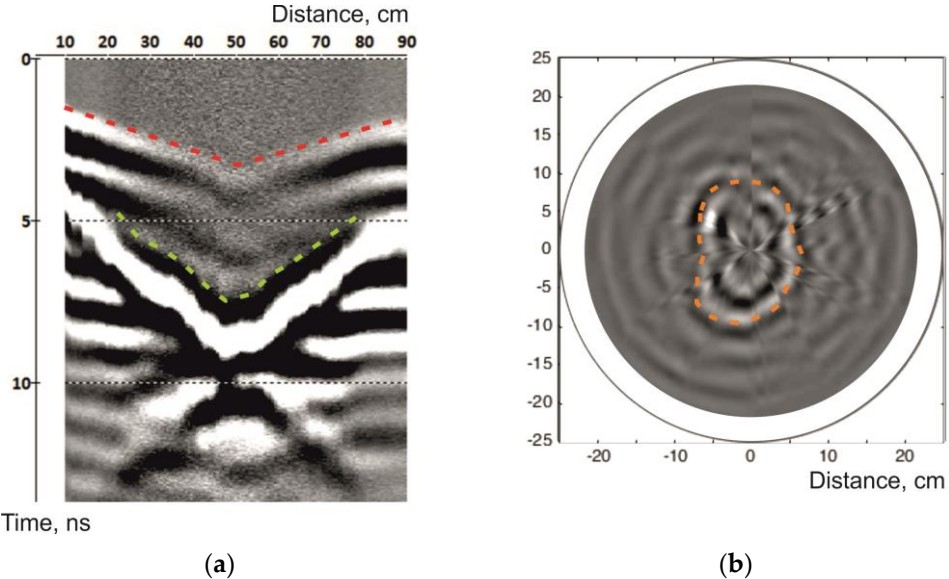

**Figure 3.** Small-leaved lime 1 GPR data: (**a**) an example of GPR tomography field data. Air wave is marked in red. The wave propagating inside the small-leaved lime is marked in green. (**b**) GPR zero-offset data in polar coordinates. Orange dotted line shows the layers boundary.

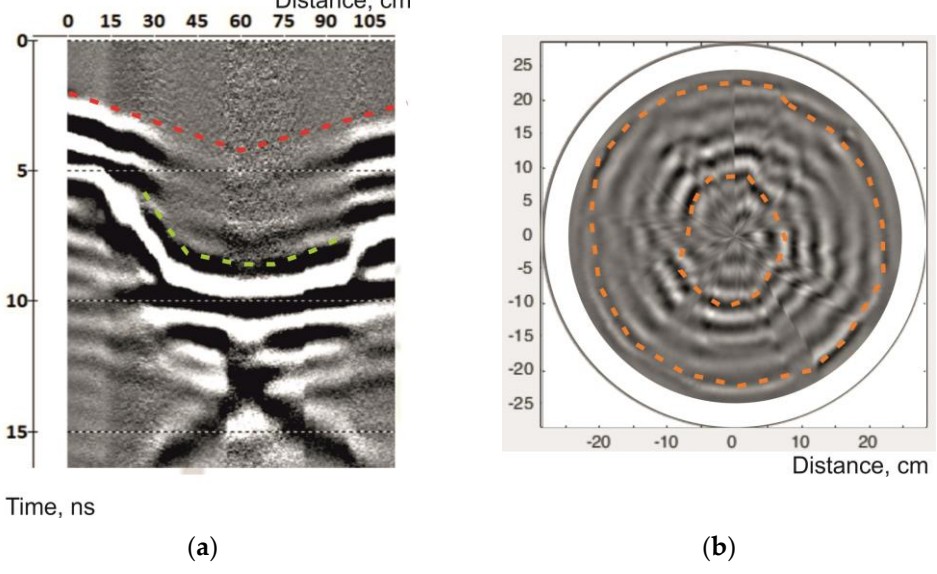

**Figure 4.** Small-leaved lime 2 GPR data: (**a**) an example of GPR tomography field data. Air wave is marked in red. The wave propagating inside the small-leaved lime is marked in green. (**b**) GPR zero-offset data in polar coordinates. Orange dotted line shows the layers boundary.

Figures 3b, 4b, 5b and 6b show examples of processed GPR data (B-Scan) obtained using zero-offset mode rearranged in polar coordinates. For time–depth conversion, the average velocity derived from tomography was used—6.4 cm/ns for small-leaved lime trees and 9 cm/ns for Scots pines.

The data obtained on small-leaved lime 1 using the ZO mode clearly show a high-amplitude reflector that divides the entire section into two circles: internal and external. The arrival time of the direct wave on the tomography data for the maximum distance between the source and the receiver (the central part of the GPR tomography data) sharply decreases. The subsidence of the hodograph of the direct wave is due to of the EM wave velocity decreasing in the central part of the small-leaved lime 1 trunk; that is, a zone of a

sharp increase in humidity in the center of the trunk. The reflection from the border of this zone can be seen in Figure 3b.

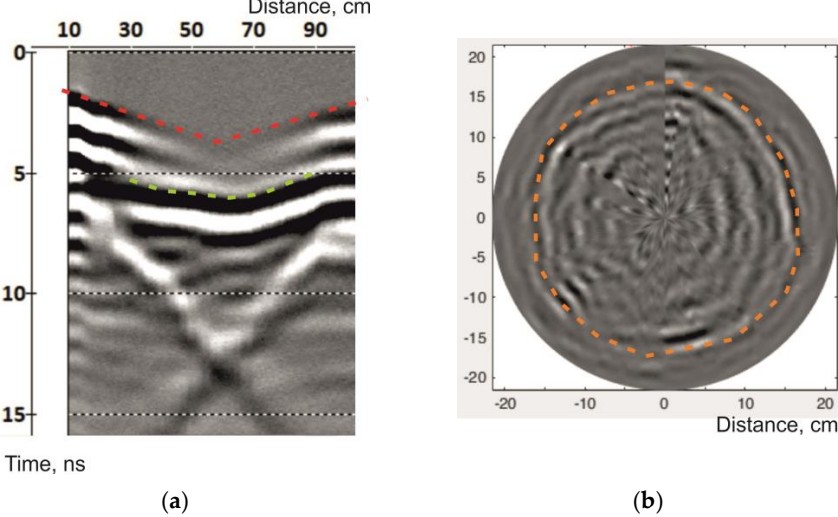

**Figure 5.** Scots pine 1 GPR data: (**a**) an example of GPR tomography field data. Air wave is marked in red. The wave propagating inside the small-leaved lime is marked in green. (**b**) GPR zero-offset data in polar coordinates. Orange dotted line shows the layers boundary.

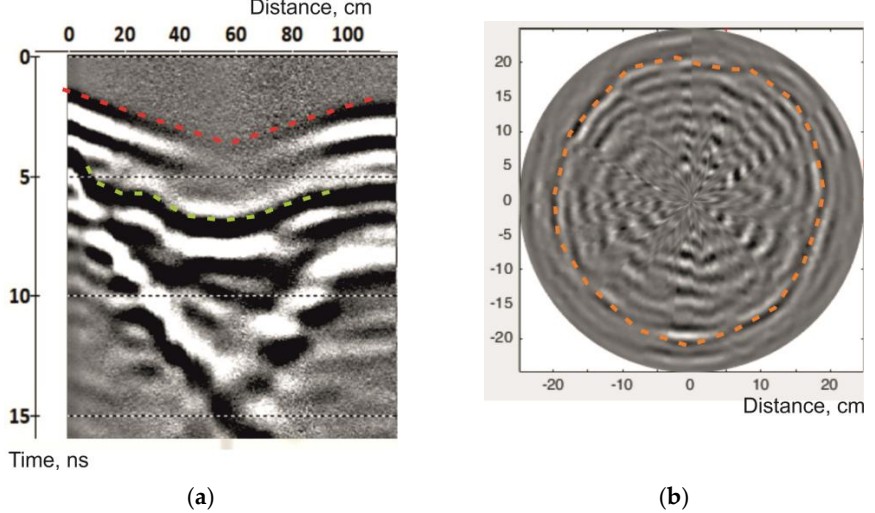

**Figure 6.** Scots pine 2 GPR data: (**a**) an example of GPR tomography field data. Air wave is marked in red. The wave propagating inside the small-leaved lime is marked in green. (**b**) GPR zero-offset data in polar coordinates. Orange dotted line shows the layers boundary.

The B-Scan from small-leaved lime 2 can be divided into three layers: the external and the internal (without reflections) and the middle one with several sub-concentric reflections. The boundaries between the zones are shown as an orange dotted line. This division reflects the three-layer structure of the small-leaved lime trunk, which is also confirmed by the results of core drilling and tomography (see below).

The direct wave arrival time changes slightly with source–receiver separation in the central part of the tomography data in Figure 4a, which indicates an increase in the velocity within the tree. That is, it defines a decrease in humidity in the central part of the trunk, opposite to small-leaved lime 1.

The direct wave arrival time in the data obtained on Scots pine trees (Figures 5a and 6a) is almost two times less than the time of arrival of the direct wave corresponding to small-leaved lime trees. This indicates less stem moisture. On the data obtained with the joint

source and receiver (Figures 5b and 6b) there are many reflections, which indicates a heterogeneous structure of the trunks for EM waves. A boundary is distinguished on Scots pine trunks at a depth of about 5–7 cm, separating the outer part without reflectors and the inner part with reflectors.

The results of GPR tomography are shown in Figures 7–10; the arrow shows the northern side of the small-leaved lime trunk. The GPR tomography results complement those obtained using ZO mode.

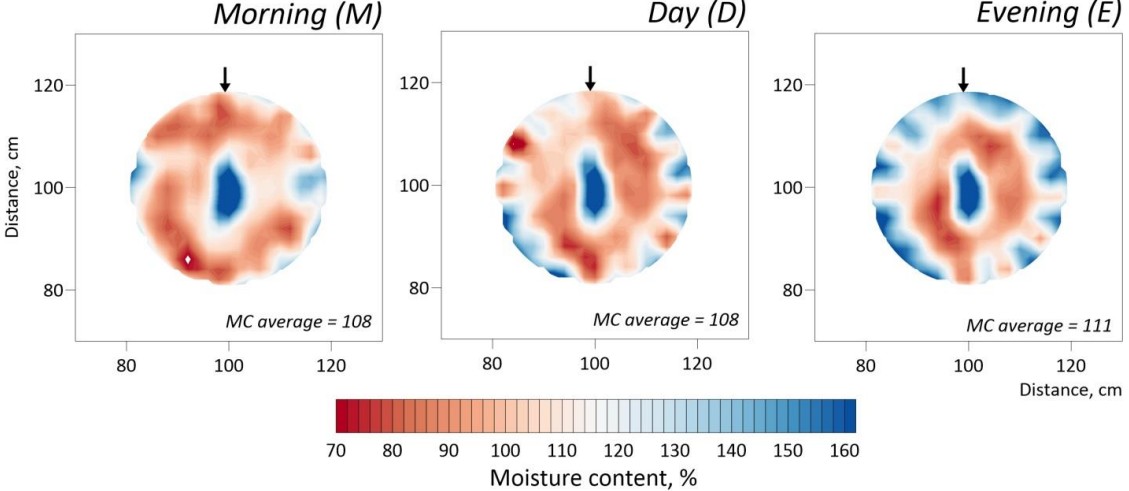

**Figure 7.** Small-leaved lime 1 GPR tomography results: MC values inside the trunk at a height of 150 cm at different times of the day.

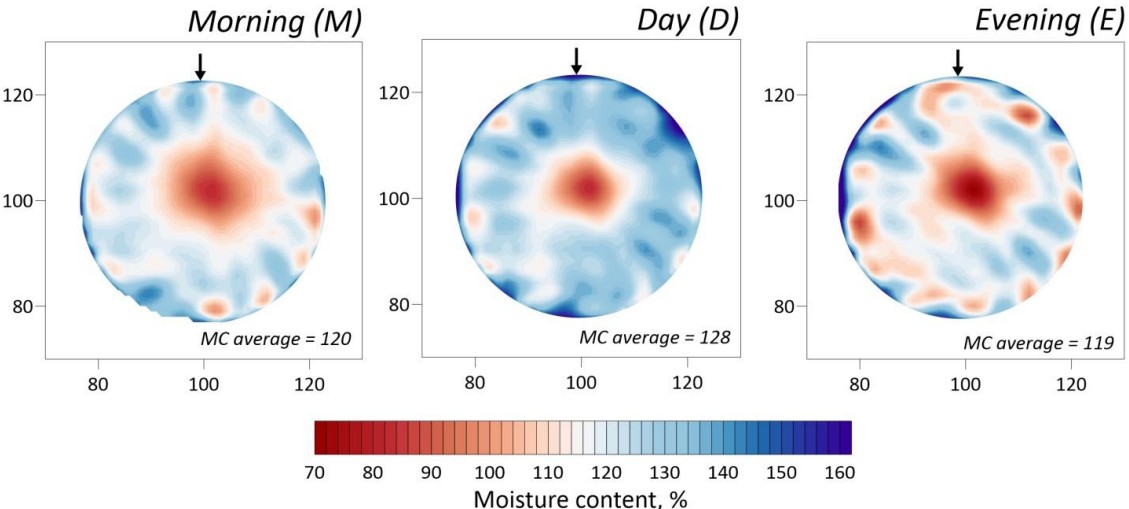

**Figure 8.** Small-leaved lime 2 GPR tomography results: MC values inside the trunk at a height of 150 cm at different times of the day.

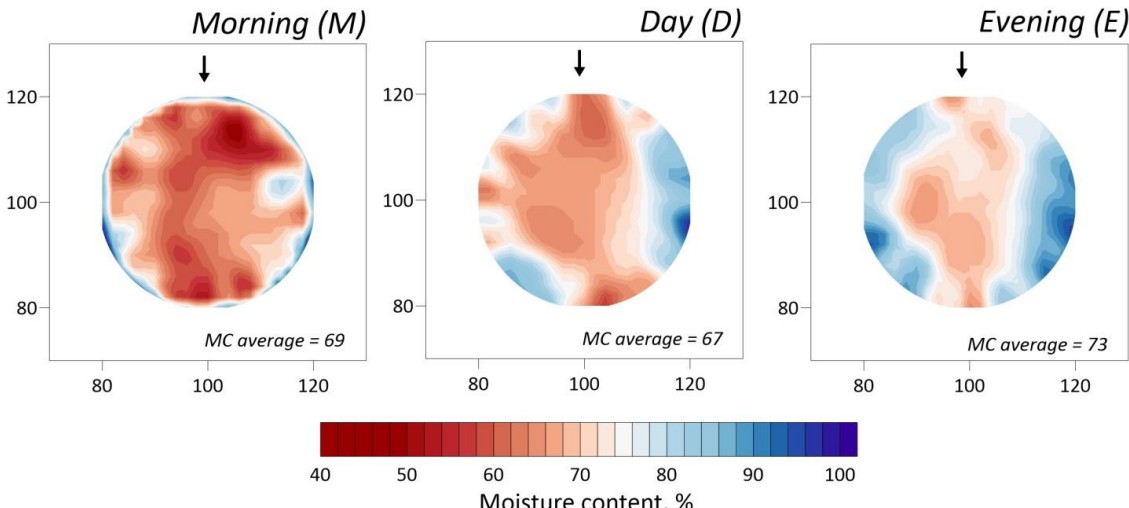

**Figure 9.** Scots pine 1 GPR tomography results: MC values inside the trunk at a height of 150 cm at different times of the day.

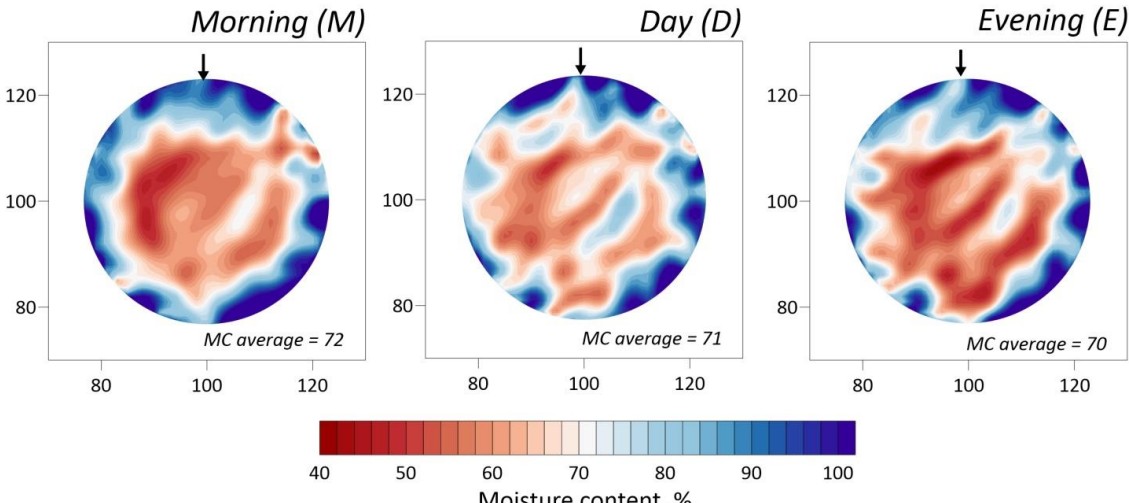

**Figure 10.** Scots pine 2 GPR tomography results: MC values inside the trunk at a height of 150 cm at different times of the day.

There is a zone of high humidity (160% and higher) in the center of small-leaved lime 1 (Figure 7). According to papers [35–37] an increase in the moisture content, porosity and dielectric constant and a decrease in the density and saturation characterize the decayed zones and development of a rotting process inside the trunk. Our measurements have no contradictions with the previous research on this subject.

On the results obtained during the day and in the evening, there is the outer layer of high humidity (about 140%–150%). On average, the moisture content of a small-leaved lime trunk is about 110%.

The average moisture content of the trunk of small-leaved lime tree 2 is 10% higher than the average moisture content of small-leaved lime tree 1 (Figures 7 and 8). Similar to the slices obtained using the zero-offset method for small-leaved lime 2, the tomographic slice of moisture distribution has also a three-layer structure. The outer layer is clearly seen only in "day" and "evening" cross-sections, with a thickness less than five centimeters. This layer is characterized by maximum MC values up to 160%. The inner "pith" of asymmetric shape, located in the center of the trunk, is characterized by minimum moisture values from 70 to 100%; its diameter is about 10 cm. The middle part, which occupies the largest

slice area, corresponds to the average moisture content for the entire trunk—about 120%. During the day, the structure does not show fundamental changes, with a slight variation of average humidity from 120% in the morning to 128% in the afternoon and back to 119% in the evening.

The average moisture content of the trunk wood of both Scots pines is about 70% (Figures 9 and 10). In the GPR tomography case, the outer wet layer is clearly distinguished inside the trunk of Scots pine 2 and is not distinguished or is distinguished only on a part of the circumference of the trunk of Scots pine 1. This result is explained by the fact that the rays passing only in this layer without leaving it are observed only at small distances between the source and the receiver when the direct signal arrivals are masked by the air wave (see Figure 5b).

The diameter of Scots pine 2 is 6.5 cm larger, so here rays that pass through the outer wet layer for a significant part of their path can be observed and their arrival could be distinguished from air wave.

## 4. Discussion

A comparison of the GPR tomography and core sampling results is shown in Figure 11. The charts of the moisture content of the trunks based on direct and indirect measurements are characterized by the same trends except Scots pine 1 trunk.

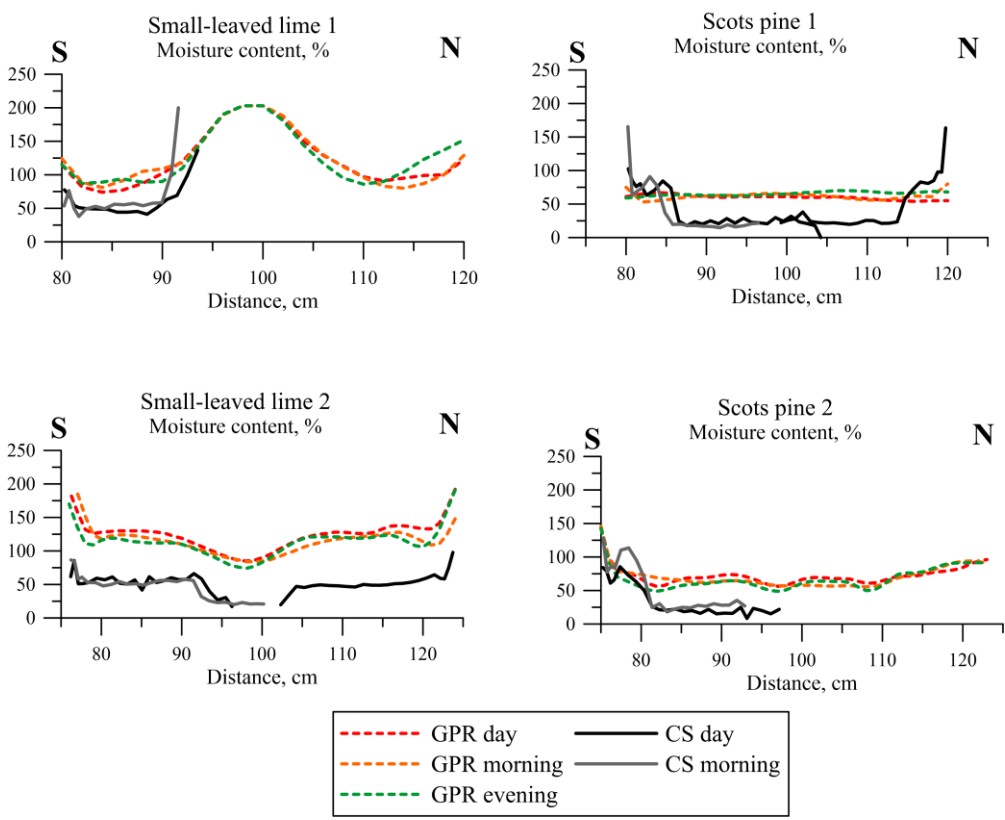

**Figure 11.** Comparison of GPR tomography and core sampling (CS) results.

During the process of core sampling of small-leaved lime 1, when the borer was plunged into the trunk at the depth of about 10–15 cm, it fell through due to low drilling resistance of the rotten area. The moisture here according to the direct data is about 200%. The rotting zone presence was confirmed by drilling, while GPR makes it possible to determine the size of this zone and its localization.

According to the results of the MC derived from cores, the small-leaved lime 2 trunk is also divided into three layers: there is a layer of maximum moisture up to 87% with a thickness of 2–3 cm directly under the bark; further into the interior of the trunk there is a

constant MC layer (about 50%). The inner layer (pith) with a diameter of about 10 cm is characterized by a minimum MC (about 25%). For small-leaved limes, all three methods show good agreement from the perspective of division into layers. According to the results of GPR and core sampling, three layers are distinguished: bark + sapwood with a thickness of less than 5 cm, heartwood with a thickness of about 15 cm and pith with a diameter of about 10 cm.

While comparing the results of indirect and direct observations for Scots pines different conclusions can be drawn. According to the results of drilling, the trunks of both Scots pines are clearly divided into two layers; the fact was observed according to GPR tomography data only for Scots pine 2. At the same time, the reflection from the bottom of this layer is visible in the data obtained by the ZO method on both Scots pines (Figures 3b and 4b).

The MC values derived from the results of tomography are two times higher than those obtained as a result of core measurements. This may be due either to the insufficient accuracy of the calculated empirical dependence or to the fact that the core was not completely dried. Meanwhile the data from both methods showed comparable spatial dynamics. Linear regression analysis showed a strong linear relationship; the coefficient of determination is more than 0.8 (Figure 12).

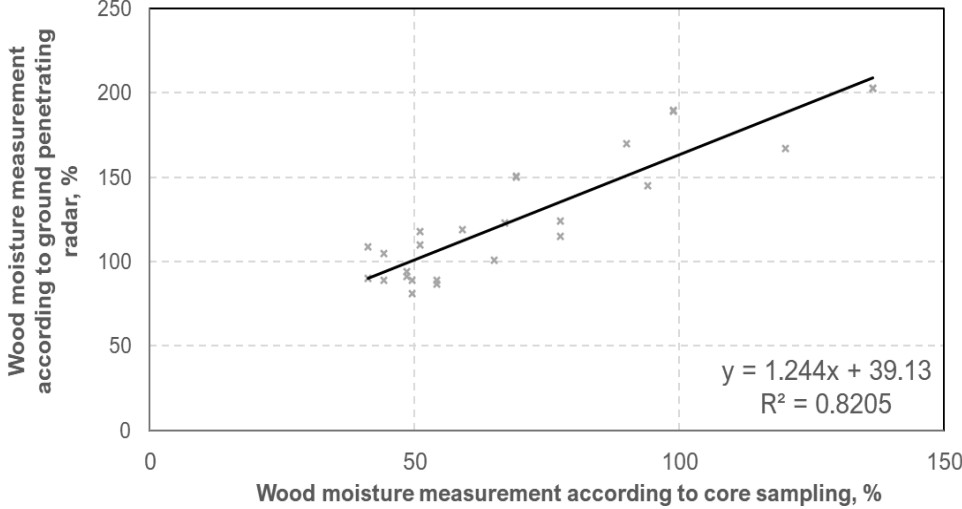

**Figure 12.** Model of linear regression between direct wood moisture measurement and GPR assessment of trunk wood moisture.

The calculated values of dielectric permittivity/moisture content are close to those cited in other papers [37–39]. Similar moisture content values are given in other papers. For example, MC of sound wood of plane trees (Platanus hybrida) reported in [37] is about 100%, while for decayed wood it ranges from 100 to 400%.

Interestingly enough, our findings show correlation with the findings in paper [40] focused on direct moisture measurements conducted on Scots pine. The authors suggested that average wood MC value decreases from 111% (40-year-old trees) to 77% (145-year-old trees). Their findings demonstrated slight changes of Scots pine heartwood MC during the year: 30%–41%, while sapwood moisture content showed no reference to the age of the tree and varied from 113% (in the summer) to 130% (in the winter) [40].

The main results of monitoring are showed in Table 1. According to the results of direct and indirect measurements, the MC changed slightly during the entire observation period. The average value of moisture content, minimum and maximum remained constant within 10% in small-leaved lime.

**Table 1.** MC in small-leaved limes (*Tilia cordata*) and Scots pines (*Pinus sylvestris*) by GPR tomography and core samples measurements (in %).

| | Small-leaved lime (*Tilia cordata*) 1 | | | | | |
|---|---|---|---|---|---|---|
| | Core Sampling | | | GPR tomography | | |
| | Morning S | Day S | Day N | Morning | Day | Evening |
| Min | 38 | 41 | 80 | 74 | 86 | 38 |
| Max | 200 | 136 | 203 | 203 | 203 | 200 |
| Average | 55 | 51 | 112 | 108 | 108 | 111 |
| | Small-leaved lime (*Tilia cordata*) 2 | | | | | |
| | Core Sampling | | | GPR tomography | | |
| | Morning S | Day S | Day N | Morning | Day | Evening |
| Min | 17 | 20 | 20 | 84 | 85 | 74 |
| Max | 86 | 87 | 98 | 187 | 192 | 191 |
| Average | 56 | 51 | 49 | 120 | 128 | 119 |
| | Scots pine (*Pinus sylvestris*) 1 | | | | | |
| | Core Sampling | | | GPR tomography | | |
| | Morning S | Day S | Day N | Morning | Day | Evening |
| Min | 15 | 10 | 20 | 53 | 54 | 59 |
| Max | 166 | 103 | 164 | 80 | 67 | 70 |
| Average | 20 | 25 | 25 | 69 | 67 | 73 |
| | Scots pine (*Pinus sylvestris*) 2 | | | | | |
| | Core Sampling | | | GPR tomography | | |
| | Morning S | Day S | Day N | Morning | Day | Evening |
| Min | 8 | 18 | 56 | 56 | 49 | 8 |
| Max | 86 | 128 | 146 | 144 | 143 | 86 |
| Average | 22 | 28 | 65 | 72 | 71 | 70 |

## 5. Conclusions

A good agreement was observed between GPR-derived moisture content and core sample-derived values. Notwithstanding GPR-derived moisture content is about two times higher than core sample-derived values, a strong linear relation with a determination coefficient more than 0.8 was observed.

A rotted area was found inside one of the small-leaved lime trees using GPR. The presence of a rotting zone was confirmed by drilling, while ground penetrating radar makes it possible to determine the size of this zone and its localization.

According to the results of measurements, a layered structure of the trunk of healthy small-leaved lime was determined: bark + sapwood less than 5 cm thick, heartwood about 15 cm thick and pith with a diameter of about 10 cm.

Moisture content of Scots pine trunks is about two times less than that of small-leaved lime trunks. For a Scots pine with a larger circumference (157 cm), it was possible to determine the two-layer structure using all three methods: outer sapwood about 5–7 cm thick and inner heartwood. For Scots pine with a smaller circumference (136 cm), the two-layer structure was determined by core sampling and ZO mode GPR data and was not clearly observed on GPR tomography results. Combination of two GPR modes can give more robust results for the discrimination of tree trunk layers.

Diurnal monitoring did not reveal any significant changes in moisture content inside the trunks.

**Author Contributions:** Conceptualization, M.S., E.T., A.K., I.S. and A.Y.; methodology, M.S. and A.Y.; validation, E.T. and A.K.; investigation, M.S., E.T., A.K., I.S. and A.Y.; writing—original draft preparation, M.S., E.T. and A.K.; writing—review and editing, M.S., E.T. and A.Y.; visualization, M.S. and I.S.; supervision, M.S. and A.Y.; project administration, M.S.; funding acquisition, A.Y. All authors have read and agreed to the published version of the manuscript.

**Funding:** Wood core sampling and measurement was supported by the Russian Science Foundation (project # 19-77-30012). This publication has been prepared with the support of the RUDN University Strategic Academic Leadership Program.

**Data Availability Statement:** The data presented in this study are available on request from the corresponding author.

**Conflicts of Interest:** The authors declare no conflict of interest.

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
