# Peer review of "Diurnal Monitoring of Moisture Content of Scots Pine and Small-Leaved Lime Trunks Using Ground Penetrating Radar (GPR) and Increment Cores"

_forests, doi:10.3390/f14020406_

Round 1

Reviewer 1 Report

Dear Authors,

It is an interesting article, however  some details should be corrected and some major information should be added in my opinion.

I present my comments in synthetic form.

Substantive comments:

The scope of the tests covers only 4 trees (2 Scots pine and 2 small-leaved lime trees) and on this basis, it is not justified to draw too far-reaching conclusions The sentence contained in Abstract and Conlusions: It can be concluded that the period of early autumn in the Moscow Region is characterized by a constant moisture content of the linden trunk during the day is invalid.   In the context of the made research, a lot of important information is missing. The description of the research object lacks information on the type of forest habitat and stand parameters. The coordinates of the forest area from which the trees for the study were selected are not given. The biosocial position of the examined trees in the stand was not given either. The article should be supplemented with information from the nearest meteorological station showing air parameters and the weather (temperature, relative humidity, cloudiness, wind force) during the period of field research. All of the above factors affect the humidity of trees (wood moisture content in their trunks) and without them the value of the obtained results is small. The ambient temperature also affects the operation of the GPR Zond-12e. Therefore, there should be information about it.    

The English name of Pinus sylvestris L. is Scots pine and Tilia cordata Mill. is small-leaved lime. This is also indicated by the standard EN 13556:2003 “Round and sawn timber - Nomenclature of timbers used in Europe”.

It is also standard to provide DBH (diameter at breast height) for forest trees at a height of 130 cm. The article gives the circumference of the trunk at a height of 150 cm.

The photo in figure 1b should be corrected (this one is not very professional: most of the photo field is a gray background, an illegible piece of paper with a handwritten number. Here should be a photo of all core samples arranged side by side with engineering length dimensioning (rules of technical drawing) or against the background of a grid with a scale on the x axis.   The difference in the obtained moisture content of wood with the use of radar and the drying-weighing method with core samples is disturbing. Probably this is due to some statistical error. The results from the core samples seem to be correct, taking into account the actual moisture content (data available in other publications) of the wood and the differences between sapwood and heartwood. Perhaps the overestimated moisture content obtained in the GPR tests results from the incorrect level of assumed wood density (500 kg/m3) in living trees it is much higher and for Scots pine and small-leaved lime it is definitely over 700 kg/3.

Editing corrections:

Due to the substantive comments described above, indicating specific places for corrections at this stage is pointless and difficult, also due to the numbering of the lines of the text is interrupted from page 3 to 11.

Here I will only point out the need for a more detailed indication of the Authors' contribution in accordance with the Instructions for Authors:

https://www.mdpi.com/journal/forests/instructions#preparation

Yours sincerely
Reviewer

Author Response

Dear Sir/Madame,

Please find attached the revised manuscript.

On behalf of my coauthors, I would like to thank you for the opportunity to revise and resubmit our manuscript (forests-2152475), entitled “Diurnal monitoring of moisture content of Scots pines and  small-leaved limes trunks using ground penetrating radar (GPR) and drilling” by Sudakova et al.

We found the reviewers’ comments to be helpful in revising the manuscript and have considered and responded to each suggestion (please find enclosed our response to reviewers).

We are sorry for the problem with the lines numbering in the previous manuscript version and have carefully checked it in the current version.

Thank you again for your consideration of our revised manuscript.

I hope you will find the revised manuscript suitable for publication in FORESTS.

Sincerely, 

Maria Sudakova

Substantive comments:

Point 1:  The scope of the tests covers only 4 trees (2 Scots pine and 2 small-leaved lime trees) and on this basis, it is not justified to draw too far-reaching conclusions The sentence contained in Abstract and Conlusions: “It can be concluded that the period of early autumn in the Moscow Region is characterized by a constant moisture content of the linden trunk during the day” is invalid. 

Response 1: The phrase was deleted.

Point 2:  In the context of the made research, a lot of important information is missing. The description of the research object lacks information on the type of forest habitat and stand parameters. The coordinates of the forest area from which the trees for the study were selected are not given. The biosocial position of the examined trees in the stand was not given either. The article should be supplemented with information from the nearest meteorological station showing air parameters and the weather (temperature, relative humidity, cloudiness, wind force) during the period of field research. All of the above factors affect the humidity of trees (wood moisture content in their trunks) and without them the value of the obtained results is small.

Response 2: The requested information was added in Materials and Methods section.

 Point 3:  The ambient temperature also affects the operation of the GPR

Response 3: During field experiment the ambient temperature was about 18 °C, so there was not any temperature influence on GPR work.

Point 4:  The English name of Pinus sylvestris L. is Scots pine and Tilia cordata Mill. is small-leaved lime. This is also indicated by the standard EN 13556:2003 “Round and sawn timber - Nomenclature of timbers used in Europe”.

Response 4: Corrected.

Point 5:  It is also standard to provide DBH (diameter at breast height) for forest trees at a height of 130 cm. The article gives the circumference of the trunk at a height of 150 cm.

Response 5: Actually we provided all measurements at height of about 130 cm (see Figure 2). Here number 150 is a mistake.

Point 6:  The photo in figure 1b should be corrected (this one is not very professional: most of the photo field is a gray background, an illegible piece of paper with a handwritten number. Here should be a photo of all core samples arranged side by side with engineering length dimensioning (rules of technical drawing) or against the background of a grid with a scale on the x axis.

Response 6: Corrected. The photo was removed.

Point 7:  The difference in the obtained moisture content of wood with the use of radar and the drying-weighing method with core samples is disturbing. Probably this is due to some statistical error. The results from the core samples seem to be correct, taking into account the actual moisture content (data available in other publications) of the wood and the differences between sapwood and heartwood. 

Response 7: The moisture content calculation from GPR data is based on some assumptions, therefore it contains errors and inaccuracies. However, the obtained moisture values do not contradict those available in the literature.Some comparisons were added in Discussion section. 

Point 8:  Perhaps the overestimated moisture content obtained in the GPR tests results from the incorrect level of assumed wood density (500 kg/m3) in living trees it is much higher and for Scots pine and small-leaved lime it is definitely over 700 kg/3.

Response 8: For calculation MC from dielectric permittivity we assume constant density 500 kg/m3 which was taken from work [Torgovnikov G (1993) Dielectric properties of wood and wood-based materials. Berlin, Heidelberg, New-York: Spring-er-Verlag, 196.]. In this work this density corresponds oven dry linden wood.

We added this information to the article in the end of section 2.1.1.

Point 9:  Due to the substantive comments described above, indicating specific places for corrections at this stage is pointless and difficult, also due to the numbering of the lines of the text is interrupted from page 3 to 11.

Response 9: Corrected.

Point 10:  Here I will only point out the need for a more detailed indication of the Authors' contribution in accordance with the Instructions for Authors:

https://www.mdpi.com/journal/forests/instructions#preparation

Response 10: Corrected.

Reviewer 2 Report

This manuscript shows the potential of using ground penetrating radar (GPR) to compute the moisture content of the tree trunks of two different species. It is a novel idea and a non-destructive method which is important in environmental studies. However, I detected some sections in the manuscript that they require major revisions. More details are given below:

Lines 31-36: Better to add few references for these statements

Line 57: Better to provide the full name first and then the acronym “Ground Penetrating Radar”

Section 2: Do we really need a different section to describe the selected species? Maybe better to make this a subsection of the “materials and methods” section?

Line 77-78: Please rephrase this sentence

Line 91: Should it be “fig”, “fig.” or “figure”? Please check throughout the manuscript.

Lines 94-95: Can you provide more information on  the “zero-offset method (ZO)”?

Line numbering is up to line 101 (page 3). This doesn’t allow a proper review of the rest of the manuscript. Also some phrases do not sound very scientific (e.g. in page 3  “to get rid of the influence of induction and conduction currents”). Please check the grammar, syntax, and line numbering.

Section 3.1.1: Please provide a detailed description of the procedure used to measure the moisture content.

Section 3.2: Were the samples taken from the same trees where the GPR measurements were conducted? Please provide some references for the method used to measure the water content.

Results: There is no description of the units used for the moisture content. How can we have a higher than 100% (water)? Is it relative? Also, figure 6 looks strange to me. How is it possible to have a higher moisture content in the heartwood? This figure is not cited in the text. Please check this section again

Discussion: I would expect to see a comparison of the findings of this study with other similar studies. It doesn’t look like a discussion section and most of the writings here should be in the results section.

Author Response

Dear Sir/Madame,

Please find attached the revised manuscript.

On behalf of my coauthors, I would like to thank you for the opportunity to revise and resubmit our manuscript (forests-2152475), entitled “Diurnal monitoring of moisture content of Scots pines and  small-leaved limes trunks using ground penetrating radar (GPR) and drilling” by Sudakova et al.

We found the reviewers’ comments to be helpful in revising the manuscript and have considered and responded to each suggestion (please find enclosed our response to reviewers).

We are sorry for the problem with the lines numbering in the previous manuscript version and have carefully checked it in the current version.

Thank you again for your consideration of our revised manuscript.

I hope you will find the revised manuscript suitable for publication in FORESTS.

Sincerely, 

Maria Sudakova

Point 1:  Lines 31-36: Better to add few references for these statements

Response 1: Corrected

Point 2:  Line 57: Better to provide the full name first and then the acronym “Ground Penetrating Radar”

Response 2: Corrected

Point 3:  Section 2: Do we really need a different section to describe the selected species? Maybe better to make this a subsection of the “materials and methods” section?

Response 3: Corrected

Point 4:  Line 77-78: Please rephrase this sentence

Response 4: Corrected

Point 5:  Line 91: Should it be “fig”, “fig.” or “figure”? Please check throughout the manuscript.

Response 5: Corrected

Point 6:  Lines 94-95: Can you provide more information on the “zero-offset method (ZO)”?

Response 6: Typical GPR surveys are collected in common offset (CO) or single-fold (SF) mode. CO acquisition deploys one transmitting and one receiving antenna that move together along the surface keeping a constant offset. Such configuration is also reposted as bistatic because it uses two antennas with separated transmitting and receiving functions. When a single antenna acts alternately as transmitting or receiving one (monostatic configuration) we have zero-offset conditions. In a CO survey a fixed geometry is usually applied, using not only a constant separation but also a fixed orientation between the antennas [citation from Forte, E.; Pipan, M. Review of multi offset GPR applications: Data acquisition, processing and analysis. Signal Process. 2017, 132, 210–220]. If the distance between the two antennas is negligible, this is also called zero-offset data (ZO).

Point 7:  Line numbering is up to line 101 (page 3). This doesn’t allow a proper review of the rest of the manuscript. Also some phrases do not sound very scientific (e.g. in page 3  “to get rid of the influence of induction and conduction currents”). Please check the grammar, syntax, and line numbering.

Response 7: Corrected

Point 8:  Section 3.1.1: Please provide a detailed description of the procedure used to measure the moisture content.

Response 8: The detailed description is provided in section 2.1.1 Moisture content calculation. More detailed description is provided in [Sudakova M., Terentyeva E., Kalashnikov A. Assessment of health status of tree trunks using ground penetrating radar tomography [J]. AIMS Geosciences, 2021a, 7(2): 162-179. doi: 10.3934/geosci.2021010]

Point 9:  Section 3.2: Were the samples taken from the same trees where the GPR measurements were conducted?

Response 9: The samples were taken from the same trees were the GPR measurements were conducted. This statement was added in the end of Materials and Methods section.

Point 10:  Please provide some references for the method used to measure the water content.

Response 10: Some references were added at section 2.1.1 Moisture content calculation  ([25-35]).

Point 11:  Results: There is no description of the units used for the moisture content. How can we have a higher than 100% (water)? Is it relative?

Response 11: To clarify this we slightly changed section 2.1.1

“Moisture content of wood (MC) is defined as the weight of water in wood expressed as a fraction, usually a percentage, of the weight of oven-dry wood [25].

MC=(Wtest-Wovendry)/(Wovendry)*100% (1)

Where Wtest: weight of test wood; Wovendry: weight of oven dry wood.

According to the equation 1, MC ranges from 0% for oven-dry wood and may exceed 200% of the weight of wood substance. In softwoods, the moisture content of sapwood is usually greater than that of heartwood [26]. The only direct method to determine MC in wood material is to measure the water content in a wood sample and the weight of oven-dry sample.”

Point 12:  Also, figure 6 looks strange to me. How is it possible to have a higher moisture content in the heartwood? This figure is not cited in the text. Please check this section again

Response 12: Corrected, Lines 306 – 311:

«There is a zone of high humidity 160% and higher in the centre of small-leaved lime 1 (Figure 7). According to papers [36, 37, 38] an increase of the moisture content, porosity, and dielectric constant and a decrease of the density and saturation characterize the decayed zones and development of a rotting process inside the trunk. Our measurements have no contradictions with the previous research on this subject».

Point 13:  Discussion: I would expect to see a comparison of the findings of this study with other similar studies. It doesn’t look like a discussion section and most of the writings here should be in the results section.

Response 13: Comparison with some results with references were added in discussion section.

Reviewer 3 Report

This study compared the trunk water content derived from a non-invasive method, namely ground penetrating radar (GPR), to destructive drilling on two pine (evergreen) and two linden (deciduous) trees. I think overall this is a well-conducted study, the manuscript is also well-prepared, and the results look promising. My only criticism goes to the limited sample size (only two trees per species were measured with GPR). Still, since I am not familiar with this method, and do not know the detailed technical specifications regarding GPR, it might be acceptable to have such a limited sample size (?). In short, I think the method present in this work looks promising and could be a handy tool to non-destructively monitor trunk water content, and thus provide insights into tree water use physiology. I only have some minor technical comments:

1.     Line50. “Among non-invasive methods of sap flow measurement electrical resistivity method “. This sentence is hard to follow, please revise it.

2.     Line76, not sure if ‘object’ needs a separate section, please check out the author's guidelines, I think commonly it should be a combined 'materials and methods’ section.

3.     Line85, “Linden 1 – 127 cm”, at my first glance, I thought the Linden tree had a circumference ranging from 1-127 cm. Maybe change it to something like ‘Linden1 (127 cm)’ to avoid this.

4.     “the grid cell size was 2 cm2.” Please superscript ‘2’ after cm.

5.     “with a density of 500 g/cm3.” Again, please superscript ‘3’ after cm.

6.     “length After deep freezing,”. Lowercase ‘After’.

7.     “there is an «air» wave”. «air»???

8.      Lastly, please add a figure to visualize the relationship between GPR and drilling measurements (i.e., linear regression between the two methods which I think was performed here). These are the core results, which definitely deserve a figure! Also, please consider adding metrics, besides R2, to assess the difference between the two measurements, such as mean absolute error (MAE), root mean square error (RMSE), etc. 

Author Response

Dear Sir/Madame,

Please find attached the revised manuscript.

On behalf of my coauthors, I would like to thank you for the opportunity to revise and resubmit our manuscript (forests-2152475), entitled “Diurnal monitoring of moisture content of Scots pines and  small-leaved limes trunks using ground penetrating radar (GPR) and drilling” by Sudakova et al.

We found the reviewers’ comments to be helpful in revising the manuscript and have considered and responded to each suggestion (please find enclosed our response to reviewers).

Thank you again for your consideration of our revised manuscript.

I hope you will find the revised manuscript suitable for publication in FORESTS.

Sincerely,

Maria Sudakova

Point 1.  Line50. “Among non-invasive methods of sap flow measurement electrical resistivity method “. This sentence is hard to follow, please revise it.

Response 1: Corrected

Point 2.     Line76, not sure if ‘object’ needs a separate section, please check out the author's guidelines, I think commonly it should be a combined 'materials and methods’ section.

Response 2: Corrected

Point 3.     Line85, “Linden 1 – 127 cm”, at my first glance, I thought the Linden tree had a circumference ranging from 1-127 cm. Maybe change it to something like ‘Linden1 (127 cm)’ to avoid this.

Response 3: Corrected

Point 4.     “the grid cell size was 2 cm2.” Please superscript ‘2’ after cm.

Response 4: Corrected

Point 5.     “with a density of 500 g/cm3.” Again, please superscript ‘3’ after cm.

Response 5: Corrected

Point 6.     “length After deep freezing,”. Lowercase ‘After’.

Response 6: Corrected

Point 7.     “there is an «air» wave”. «air»???

Response 7: Corrected

Point 8.      Lastly, please add a figure to visualize the relationship between GPR and drilling measurements (i.e., linear regression between the two methods which I think was performed here). These are the core results, which definitely deserve a figure! Also, please consider adding metrics, besides R2, to assess the difference between the two measurements, such as mean absolute error (MAE), root mean square error (RMSE), etc. 

Response 8: Corrected, figure 11 was added.

Round 2

Reviewer 2 Report

The authors adressed most of my comments and suggestions. I have nothing more to add. I wish them good luck!